# The Computational Acid–Base Chemistry of Hepatic Ketoacidosis

**DOI:** 10.3390/metabo13070803

**Published:** 2023-06-28

**Authors:** Samuel L. Torrens, Robert A. Robergs, Steven C. Curry, Marek Nalos

**Affiliations:** 1School of Exercise and Nutrition Sciences, Faculty of Health, Queensland University of Technology, Kelvin Grove, QLD 4058, Australia; 2Department of Medical Toxicology, Banner—University Medical Center Phoenix, Phoenix, AZ 85006, USA; 3Intensive Care Medicine, Goulburn Hospital, Goulburn, NSW 2580, Australia

**Keywords:** acid–base balance, biochemistry, ketones, ketoacidosis, metabolism

## Abstract

Opposing evidence exists for the source of the hydrogen ions (H^+^) during ketoacidosis. Organic and computational chemistry using dissociation constants and alpha equations for all pertinent ionizable metabolites were used to (1) document the atomic changes in the chemical reactions of ketogenesis and ketolysis and (2) identify the sources and quantify added fractional (~) H^+^ exchange (~H^+^e). All computations were performed for pH conditions spanning from 6.0 to 7.6. Summation of the ~H^+^e for given pH conditions for all substrates and products of each reaction of ketogenesis and ketolysis resulted in net reaction and pathway ~H^+^e coefficients, where negative revealed ~H^+^ release and positive revealed ~H^+^ uptake. Results revealed that for the liver (pH = 7.0), the net ~H^+^e for the reactions of ketogenesis ending in each of acetoacetate (AcAc), β-hydroxybutyrate (β-HB), and acetone were −0.9990, 0.0026, and 0.0000, respectively. During ketogenesis, ~H^+^ release was only evident for HMG CoA production, which is caused by hydrolysis and not ~H^+^ dissociation. Nevertheless, there is a net ~H^+^ release during ketogenesis, though this diminishes with greater proportionality of acetone production. For reactions of ketolysis in muscle (pH = 7.1) and brain (pH = 7.2), net ~H^+^ coefficients for β-HB and AcAc oxidation were −0.9649 and 0.0363 (muscle), and −0.9719 and 0.0291 (brain), respectively. The larger ~H^+^ release values for β-HB oxidation result from covalent ~H^+^ release during the oxidation–reduction. For combined ketogenesis and ketolysis, which would be the metabolic condition in vivo, the net ~H^+^ coefficient depends once again on the proportionality of the final ketone body product. For ketone body production in the liver, transference to blood, and oxidation in the brain and muscle for a ratio of 0.6:0.2:0.2 for β-HB:AcAc:acetone, the net ~H^+^e coefficients for liver ketogenesis, blood transfer, brain ketolysis, and net total (ketosis) equate to −0.1983, −0.0003, −0.2872, and −0.4858, respectively. The traditional theory of ketone bodies being metabolic acids causing systemic acidosis is incorrect. Summation of ketogenesis and ketolysis yield H^+^ coefficients that differ depending on the proportionality of ketone body production, though, in general, there is a small net H^+^ release during ketosis. Products formed during ketogenesis (HMG-CoA, acetoacetate, β-hydroxybutyrate) are created as negatively charged bases, not acids, and the final ketone body, acetone, does not have pH-dependent ionizable groups. Proton release or uptake during ketogenesis and ketolysis are predominantly caused by covalent modification, not acid dissociation/association. Ketosis (ketogenesis and ketolysis) results in a net fractional H^+^ release. The extent of this release is dependent on the final proportionality between acetoacetate, β-hydroxybutyrate, and acetone.

## 1. Introduction

There are numerous terms involved in the labeling of the increased production of ketone bodies. For clarity, in this manuscript, the production of ketone bodies by the liver is referred to as ketogenesis, which is a continuous, naturally occurring state that can vary in magnitude and severity. There are three recognized ketone bodies, acetoacetate (AcAc), β-hydroxybutyrate (β-HB), and acetone. The blood concentrations of each ketone body can be highly variable during different rates of ketogenesis (data provided are for normal vs. elevated conditions, respectively β-HB = 0.05 vs. 23 mmol/L, AcAc = 0.05 vs. 2.0 mmol/L, acetone = 0.05 vs. 11.00 mmol/L) [1,2,3]. The oxidation of ketone bodies is referred to as ketolysis, and the more general reference to elevated ketone bodies involving combined ketogenesis and ketolysis is referred to as ketosis.

The first documented discovery of ketone bodies can be traced back to the latter part of the 19th century when they were discovered in the urine samples of patients in diabetic coma [4]. Due to the limited scientific technology and understanding of that time, the associated presence of ketone bodies in urine during such dire clinical conditions with high mortality rates set strong foundations for ketone bodies to be viewed as toxic by-products of lipid oxidation. This nefarious status of ketone bodies remained for over 100 years until 1966, when Cahill et al. [5] first proposed that a benefit of ketosis could be the use of ketone bodies as an alternative to glucose to fuel brain metabolism.

During conditions of prolonged restricted cellular glucose uptake, such as diabetes mellitus or the reduced intake of carbohydrates (severe caloric restriction), the human brain transitions predominantly from glucose-derived ATP production to ketone-body-derived ATP production [3]. This critical survival process is a consequence of not only restricted cellular glucose uptake or availability but also the human brain’s inability to directly utilize fatty acids as an energy source. While ketosis occurs continuously to a small extent, even during times of fully functioning glucose uptake (plasma ketone levels 0.05–0.5 mmol/L), ketogenesis is dramatically increased during times of glucose restriction or deficiency (plasma ketone levels > 3.0 mmol/L) [3,6,7,8]. During these conditions, coincident to a decrease in glycolytic flux, is an increased mobilization and transport of fatty acids to the liver to undergo β-oxidation, thereby increasing acetyl-CoA production to such an extent that excess acetyl CoA is converted to ketone bodies to prevent depletion of free CoA [3,5]. These ketone bodies can then be transported to the brain and other tissues as an alternative fuel source to glucose for oxidation and ATP production [7]. While acute, low-level elevations in ketone bodies are not considered dangerous; for example, ketogenic diets are designed specifically to induce small to moderate elevations in the blood concentrations of ketone bodies; severe ketosis can result in conditions, such as diabetic ketoacidosis (DKA), alcoholic ketoacidosis, and starvation ketoacidosis leading to hospitalization, coma, or death [9].

Prolonged high rates of ketosis are associated with systemic acidosis (currently termed ketoacidosis). As explained by Dreschfeld in 1886 [4], during this time, the acidosis of severe diabetic ketoacidosis was assumed to be cause–effect; in other words, ketone bodies were thought to be produced as metabolic acids. Such an interpretation pre-dated advances in acid–base chemistry and physiology, similar to the errors of the early understanding of the metabolic biochemistry of the cellular lactic acid production [10]. Consequently, the early 20th-century mistaken belief of a cellular lactic acid, rather than lactate, production may have reinforced the construct of ketone bodies as metabolic acids and the subsequent traditional metabolic explanation of the condition of ketoacidosis.

Each of β-HB and AcAc is chemically categorized as carboxylic acids due to their carboxylic acid functional groups and associated pKa’s of 4.41 and 3.58, respectively [11]. However, similar to the production of lactate at physiological pH with a pKa of 3.67 [11], the organic chemistry reveals that each of these “acids” is produced in their deprotonated, ionic state. Nevertheless, as with the ongoing error of the cellular lactic acid production [12,13], the chemical structural classification of ketone bodies (β-HB and AcAc) as acids has further reinforced the view that following their metabolic production, each of β-HB and AcAc will dissociate and release a proton (H^+^) within the physiological pH range (6.00–7.60). In extension, if this metabolic condition remains uncorrected, systemic acidosis can ensue.

Despite advances in acid–base chemistry and computations of the extent of H^+^ exchange during chemical reactions [14,15,16,17], the historically dependent metabolic acid interpretation of ketosis has remained up to the current time. For example, the depictions of both AcAc and β-HB in protonated form from Van Itallie [18] and Kamel and Halperin [19] clearly showed that the misinterpretation of the “acidic” nature of ketones has long been connected to the further (mis)understanding and (mis)interpretation of ketoacidosis. Furthermore, current articles on ketosis still claim without proof or reference [2,20,21] that the large hydrogen-ion dissociation of ketones leads to metabolic acidosis. Even prominent modern textbooks on biochemistry [22] that correctly depict the production of AcAc and β-HB in their ionic forms still state that it is the “acidic” nature of ketones that is the cause of the associated acidosis.

While the mortality rates and treatment of ketoacidosis have significantly improved since the late 19th century, the negatively associated stigma remains for the interpretation that ketone bodies are produced as acids that dissociate at physiological pH, resulting in systemic metabolic acidosis. Occurrences of ketoacidosis are still common in hospital critical care wards, with prevalence in undiagnosed or poorly controlled type I and II diabetes mellitus (diabetic ketoacidosis) [23], alcoholic ketoacidosis [24], and starvation ketoacidosis associated with eating disorders [25]. It is logical and rational to have the view that the source of the underlying acidosis should be known for the most effective preventative treatment methods to be developed. Yet, this is not possible unless the underlying source of the acidosis is understood. This is problematic because the probability of improved treatment options for disease state systemic ketoacidosis in critical care settings would likely benefit greatly if the true metabolic cause of the acidosis is known.

Given the sustained disagreement and differing interpretations of H^+^ exchange during the cellular production of ketone bodies and the associated metabolic acidosis, the purpose of this research was to (a) present the charge and H^+^ balanced chemical reactions of ketone body production (ketogenesis), (b) provide the dissociation constants for these reactions and compute the data for the H^+^ exchange for each ketone body across the physiological pH range (severe cellular metabolic acidosis to systemic alkalosis; pH = 6 to 7.6), (c) perform the above for the oxidation of ketone bodies (ketolysis) that occurs in peripheral tissues (brain and skeletal muscle), and (d) provide recommendations for future research to inform metabolic explanations for the systemic acidosis that coincides with ketosis.

## 2. Materials and Methods

### 2.1. Data Reference Sources

Substrates and products, their chemical structures, and the organic chemistry of the atomic changes during the chemical reactions involved in ketogenesis and ketolysis were obtained from textbooks on metabolic biochemistry [22] and adjusted where necessary based on prior errors in balancing hydrogen atoms across substrates and products (Figure 1 and Figure 2; Table 1). Data for the pKa values of each metabolite were obtained from the purchase of a commercially available pKa calculator (Chemaxon) (Table 1) [11]. Data reference sources for the pH of bodily compartments were obtained from the following: brain [26]; skeletal muscle [27]; blood [22]; and liver [28].

### 2.2. Computations

Computation of the extent of H^+^ exchange via covalent modification and association/dissociation of each metabolite of ketogenesis and ketolysis were performed as previously explained by Robergs [14] for pH conditions spanning 6.0 to 7.6. For example, once dissociation constants were retrieved for specific metabolites of the reactions of ketogenesis and ketolysis, a spreadsheet (Microsoft^®^ Excel^®^ for Microsoft 365, version 2301, Microsoft Corporation, Redmond, Washington, DC, USA) content was devised to calculate H^+^-bound and unbound fractions of each metabolite using an alpha equation (Equation (1)) for pH conditions from 6.0 to 7.6 in 0.1 pH unit increments. The free ligand (L^−^) and H^+^ bound (L-H) fractions of the metabolite could then be solved.
(1)$$\alpha_{-n}=\frac{\left[L^{n-}\right]}{L_{tot}}=\frac{1}{1+K_{H}\left[H^{+}\right]}$$
 L = ligand (molecule); K = dissociation constant.

The H^+^ free and bound fractions of the metabolites across the pH range were then compiled into specific reaction expressions for the conversion of substrates to products. The best example of this process is for the β-HB Dehydrogenase reaction, which involves both pH-dependent H^+^ exchange and covalent H^+^ uptake. The difference between the H^+^ bound sum of the metabolites of the products to that of the substrates, plus the covalent H^+^ uptake, equaled the H^+^ coefficient for the reaction at each given pH value. Reference conditions for pH for the liver, muscle, and brain were 7.0, 7.1, and 7.2, respectively. These conditions were used to retrieve pH-specific reference data for H^+^ coefficients for ketogenesis (liver at pH = 7.0) and ketolysis (muscle at pH = 7.1; brain at pH = 7.2), based on known ratios of products from ketogenesis, as documented in prior research [29,30,31]. It was assumed that ketolysis (oxidation) of β-HB and AcAc was shared equally (50%-50%) between muscle and brain. As there is limited biochemical and prior research evidence of the proportionality between the metabolic vs. lung clearance of the acetone [29,30,31], we did not account for the oxidation of acetone in calculations of the H^+^ coefficient of ketolysis. Furthermore, due to the limited research involving radioisotope labeling of atoms in ketone bodies to quantify rates of appearance and clearance, data on concentrations of ketone bodies in blood were used to estimate differences in the cellular production of each ketone body.

Once the computations were completed for all reactions of ketogenesis and ketolysis (Figure 1 and Figure 2), further calculations were structured into three distinct phases (liver ketogenesis, blood transfer, and muscle and brain ketolysis) to account for the changing pH conditions of each bodily compartment. Summation of the H^+^ exchange for given pH conditions for all substrates and products of each reaction of a ketone body resulted in a net pathway H^+^ exchange coefficient, where negative revealed H^+^ release and positive revealed H^+^ uptake. Tissue concentrations of all three ketone bodies vary considerably between individuals and based on the severity and duration of the ketosis [29,30,31]. Consequently, for clarity in comparisons, variation in the proportionality of all three ketone bodies was used to compare net ~H^+^e coefficients resulting from liver ketogenesis (pH = 7.0), transfer of ketone bodies to blood (pH = 7.4), and the reactions of ketolysis in muscle and brain tissue (pH = 7.1 and 7.2 for muscle and brain, respectively).

## 3. Results

As previously explained, results for the chemical structures of substrates and products of the reactions of ketogenesis and ketolysis are presented in Figure 1 and Figure 2. Pertinent dissociation constants are presented in Table 1, and Table 2 presents data calculated for the H^+^ coefficients of the substrate and products and net reaction H^+^ exchange of the β-HB Dehydrogenase reaction across the physiological pH range.

Figure 3 presents the pH-dependent change in ionization of the three main ionizable substrates and products involved in ketogenesis. These data are the result of the alpha equations (Equation (1)) applied to the specific pH condition data for each metabolite. Table 3 presents the net H^+^ coefficients for all the reactions of ketogenesis and ketolysis for pH conditions of 6.0, 7.0, and 7.6. Table 4 presents the net sum of all reactions of ketosis (liver ketogenesis + blood transfer + brain ketolysis) to reveal a net ketosis H^+^ coefficient based on different proportionalities of the final concentrations of the three ketone bodies.

Despite acetone production rarely being included in prior research and commentary on ketoacidosis, our data show that it is important to include the production of acetone because the production of each β-HB and acetone from AcAc consumes close to 1 H^+^ (Figure 1). As such, the greater the representation of β-HB and acetone as ketone bodies, the greater the metabolic buffering of the H^+^ release from the HMG-CoA reaction during ketogenesis. Furthermore, the greater the proportion of acetone production during ketosis, the less the capacity for ~H^+^ release from the β-HB Dehydrogenase reaction during ketolysis (Table 4). These realities differ from the mainstream clinical, biochemical interpretation of the biochemistry of ketoacidosis.

## 4. Discussion

In much the same way as computational chemistry models contributed fundamental evidence to the understanding of H^+^ exchange in skeletal muscle during intense exercise [10,14,15,16], the current investigation used computational chemistry modeling to provide further insight into ~H^+^e during the process of ketosis (ketogenesis, uptake into different tissues, and ketolysis). Traditional interpretations of ketosis involve the view that ketone bodies are produced as fully associated metabolic acids that dissociate upon production, resulting in H^+^ release and a gradual H^+^ accumulation that leads to systemic acidosis, currently referred to as ketoacidosis [22]. More recently, these views have been challenged, causing conjecture within the medical and academic communities [17]. The current debate surrounding ~H^+^e during the condition of ketoacidosis primarily revolves around two points of contention. Firstly, the primary source of ~H^+^ release/accumulation that leads to clinically significant pH changes during prolonged high rates of ketosis [17], and secondly, the mechanism of action that is responsible for the ~H^+^ release/accumulation [10,14,15,16].

Our results from the computational chemistry of ~H^+^e for all substrates and products indicate that meaningful net ~H^+^ release occurs during ketosis. Of the four reactions of β-HB and acetone production and the three reactions of AcAc production during ketogenesis, calculations show that H^+^ release occurs during the HMG-CoA Synthase reaction (Figure 1b). Added H^+^ release also occurs during ketolysis via the reversal of the β-HB Dehydrogenase reaction (Figure 2a). Such ~H^+^ release is opposed by ~H^+^ uptake of the β-HB Dehydrogenase and AcAc Decarboxylase reactions of ketogenesis. All remaining reactions are negligible in their contribution to ~H^+^e (Figure 1 and Figure 2). These calculations indicate that a meaningful increase in rates of ketosis (such as during times of starvation or poorly controlled diabetes) could theoretically contribute to an increase in H^+^ release and subsequent systemic acidosis, currently termed ketoacidosis. However, it must be stated that meaningful increases in rates of ketosis are also accompanied by many additional alterations of cellular metabolism spanning multiple tissues. These alterations include but are not limited to, increased rates of lipolysis, changes in plasma ketone ratios, changes in uptake and oxidation rates of ketone bodies in multiple tissues, and alterations to glycolysis and β-oxidation pathways. These alterations have previously been documented in the early works of Cahill [5] and more recently explored by Green and Bishop [17]. Consequently, the results of this research only address our stated purpose, which was to evaluate the reactions of ketogenesis and ketolysis to discern their roles as a source of the ~H^+^e during ketosis.

While the results of this research support the acidic nature of ketosis, the mechanism by which this net H^+^ release occurs does not support traditional views. As previously stated, ~H^+^ release occurs during the HMG-CoA Synthase reaction of ketogenesis (Figure 1b) and the β-HB Dehydrogenase reaction of ketolysis (Figure 2a). In reference to traditional views, this indicates that meaningful ~H^+^ release occurs prior to the creation of any ketone body during ketogenesis and during the catabolism (ketolysis) of β-HB. As such, the traditional theory of the “acidic nature” of ketone bodies causing systemic acidosis is not plausible. For traditionalists who would argue that the metabolic intermediates of ketone body production during ketogenesis, rather than the ketone bodies themselves, are acidic in nature, calculations show that no reactions of ketogenesis result in pH-dependent H^+^ dissociation of acidic functional groups (Figure 1).

The H^+^ release that occurs during the HMG-CoA Synthase reaction of ketogenesis occurs when water is covalently modified to release an H^+^ after the addition of a hydroxyl group to HMG-CoA (hydrolysis) (Figure 1b). Our calculations show that such additions occur via covalent modification and not pH-dependent ionization, which, therefore, explains the near-constant profile of ~H^+^e shown in Figure 3. For example, compare this result to that of the metabolites and reactions of glycolysis, as documented by Robergs [14]. In a similar manner, and a fundamental finding of this research, metabolic H^+^ release occurs via other chemical processes and not through the generation of organic acids. This understanding is also relevant during ketolysis for the ~H^+^ release for the reversal of the β-HB Dehydrogenase reaction causing the conversion of β-HB to AcAc (Figure 2a). Significant H^+^ release does not occur via acid dissociation but instead by the oxidation–reduction nature of the reaction. If we are to continue accepting the narrative of traditional views by which the “acidic nature” of ketones is the cause of ketoacidosis while clearly demonstrating that H^+^ release is a result of covalent modifications, then for consistency, these same views must be applied across the biological spectrum to all organic reactions. However, if these views are applied across the biological spectrum, it becomes clear that their application is severely inappropriate. For example, substrates, such as ATP, would need to be classified as a metabolic acid, as hydrolysis of ATP during the reaction ATPase produces ADP + Pi and releases an H^+^. This is the same covalent modification reaction (hydrolysis) as during the HMG-CoA Synthase reaction of ketogenesis, yet the traditionalist interpretation classifies ketone bodies as acids but not ATP. Depending on the differences between the dissociation constants of substrates vs. products, added minor H^+^ exchange can occur via pH-dependent association and/or dissociation, though once again, note that for ketosis, there is minimal pH-dependent dissociation or association.

Since their discovery over 100 years ago, the acidic nature of ketone bodies has been continually reinforced up to the present time [1,2,22]. In stark contrast to historical and traditional views stating ketone bodies are responsible for the acidosis associated with severe ketosis, H^+^ exchange computations of β-HB (Table 2) show that β-HB is a net H^+^ consumer during ketogenesis. This finding is a glaring reality check for the present time, where the understanding, interpretation, and research inquiry into ketone bodies and the process of ketosis has been in error. Computations of ~H^+^e show that AcAc is the only ketone body in which the production results in net (across all reactions leading to production) H^+^ release. In other words, both β-HB and acetone production buffer against ~H release (Figure 1, Table 2).

Similar to the increasing presence of lactate during anaerobic glycolysis, β-HB is an H^+^ consumer/buffer against acidosis that can also be used as a substrate for ATP production. It is also important to re-emphasize that H^+^ consumption during the conversion of AcAc to β-HB is not a result of acid association. It is the result of the oxidation–reduction reaction where NADH is oxidized to NAD^+^, and AcAc is reduced to β-HB. This would make physiological sense for why the ratio of β-HB to AcAc in the blood increases from 1:1 to 10:1 [30] during severe ketosis, presumably in compensation for the significant net H^+^ release when AcAc is not converted to β-HB (Table 4). This finding is not only significant in the further understanding of acid–base chemistry for organic reactions but may, in fact, have tangible benefits in the treatment of ketoacidosis. Currently, increasing concentrations of plasma ketone bodies are considered a concern for acid–base balance; however, a more appropriate focus may be on the plasma concentration of AcAc, as it is when AcAc is not converted to β-HB or acetone that a larger net ~H^+^ release would occur during ketogenesis.

It is commonly noted in research papers, textbooks, and by medical professionals that the “fruity” breath of individuals during ketoacidosis is caused by excessive blood acetone and is a key indicator of the condition. Yet, acetone’s influence on acid–base balance during ketoacidosis is often neglected and under-reported, which is likely due to its historical perception as a harmful by-product of ketosis due to its volatile chemistry and expulsion from the body via urine and exhaled air [29,30,31]. Structurally, acetone is neither an acid nor a base, which may further explain why it has been underrepresented, as it does not fit the historical narrative that the acidic nature of ketone bodies causes ketoacidosis. In contrast, the influence of acetone is an important consideration for acid–base balance during ketosis, as the ~H^+^e coefficients during the conversion of AcAc to acetone show meaningful ~H^+^ consumption (Figure 2, Table 4). It has been previously shown that during starvation ketoacidosis, blood concentrations of acetone can be similar [2] or in excess of AcAc [29,30,31]. Our results have used this data to reveal how the production of acetone during ketosis plays a large role in reducing the net ~H^+^ release and, as such, is an extremely important H^+^ consumer during high rates of ketosis (Table 4). Nevertheless, the role of the metabolic removal of acetone is difficult to estimate at this time due to limited evidence for the chemical reactions involved and the exact fraction of the acetone pool that is cleared in this manner [29,31].

When trying to elucidate the importance of ketosis to systemic acid–base balance, it is important to consider the systematic changes in ~H^+^e that may occur when transitioning from glucose-derived ATP production to ketone-derived ATP production. Ketone bodies replace the utilization of glucose for cellular ATP production, so it is an important consideration to determine the net H^+^ release when comparing the catabolism of both substrates. In simple terms, would ketolysis contribute more, less, or the same amount of H^+^ to the system than that of the process it replaces (glycolysis)? Previous research of the H^+^ exchange during glycolysis from Robergs [10] indicates that the H^+^ exchange from glycogenolysis/glycolysis would be greater than or equal to that of ketolysis, which, of course, leaves the evidence of a severe systemic blood acidosis during ketosis to require further open-minded research inquiry and explanation.

### Limitations

It is openly acknowledged that the calculations and factors used in the current investigation are not exhaustive in their findings, and further research is required to account for additional factors during ketosis. These include but are not limited to, competing cations during all phases of ketosis, the varying pH levels of different bodily compartments during ketolysis in addition to the brain and skeletal muscle, systemic changes in pH during the development of ketoacidosis (i.e., the role of the MCTs in metabolite movement in and out of cells, increased rates of lipolysis, decreased glycolytic flux), changes in the ratio of ketone bodies and the varying uptake rates of ketone bodies from different tissues through the entire development of ketoacidosis. It is also important to note that tissue concentrations of the ketone bodies are unlikely to be true to the cellular production rates, especially when considering the simultaneous functioning of ketogenesis and ketolysis. Similarly, the complications surrounding the uncertain metabolic clearance of acetone require future measurements and clarification, and it would be pertinent to replicate these procedures using in vitro preparations of metabolic pathways.

The data presented here are specifically for the H^+^ exchange coefficients for the chemical reactions of ketogenesis and ketolysis, which directly address the purpose of this research, which was to test whether the production of ketone bodies directly causes ~H^+^ release, and, if so, for what reason. Further research is needed that adds the reactions involved in the metabolic perturbations that accompany ketosis.

## 5. Conclusions

The process of ketosis is made up of two phases, ketogenesis, and ketolysis. Ketogenesis produces a mix of AcAc, β-HB, and acetone. The HMG CoA Synthase reaction releases close to 1 H^+^ in the pathway to AcAc production. AcAc can be converted to β-HB or acetone, and both reactions consume ~H^+^. The proportionality of AcAc:β-HB:acetone will determine the extent of the ~H^+^ uptake, with greater acetone production raising the ~H^+^ uptake. Once ketone bodies have undergone ketogenesis in the liver, they are then transported to tissues around the body to undergo ketolysis and subsequent utilization for ATP production. Ketolysis is not simply a reversal of the reactions of ketogenesis but a series of three reactions, whereby β-Ketoacyl-CoA Transferase replaces the HMG-CoA Synthase and HMG-CoA Lyase reactions of ketogenesis. Meaningful ~H^+^ release occurs during ketolysis through the reversal of the β-HB Dehydrogenase reaction.

Results of computational chemistry show the plausibility of an H^+^ accumulation during the process of ketosis; however, the mechanism of this H^+^ accumulation is revealed to be via covalent modification, not acid dissociation. These findings add further weight to the increasing concern that traditional views on acid–base chemistry, when applied to biological reactions (in this case, ketoacidosis), are inappropriate. The findings support more modern views that cells do not produce metabolic acids but, rather, negatively charged bases. Recent research inquiry into traditional views of ketoacidosis has unveiled multiple flaws, inconsistencies, and claims that lack evidence-based scientific inquiry. Future research inquiry should extend this research to include added metabolic reactions that occur coincident with ketosis, improved understanding of the clearance of acetone in the body, and added inquiry to modified treatments that better target the reactions directly involved in ~H^+^ release during ketosis.

## Figures and Tables

**Figure 1 metabolites-13-00803-f001:**
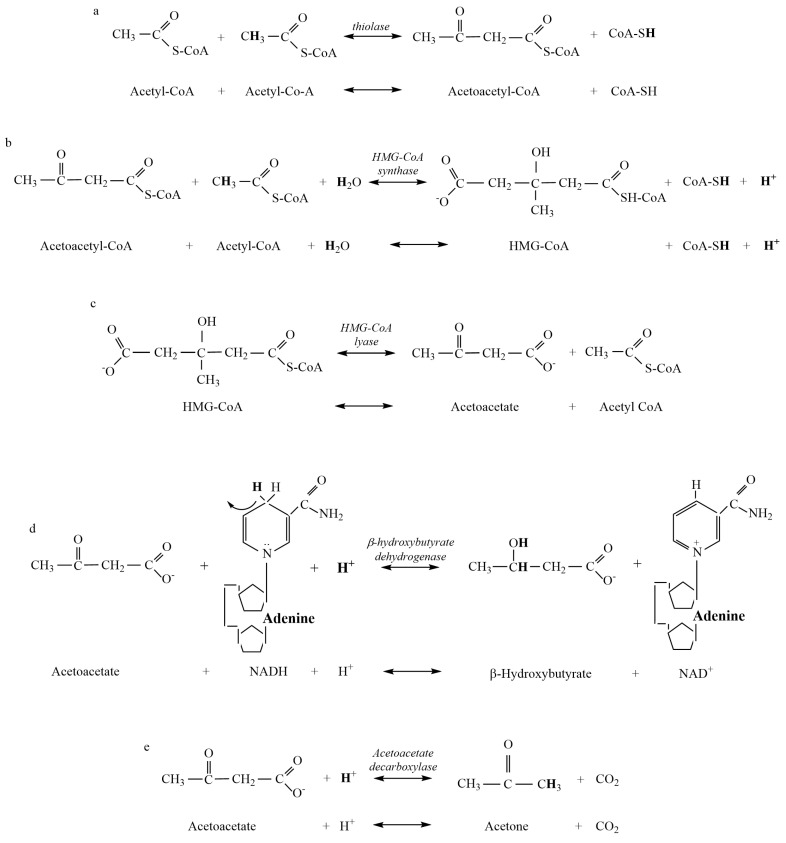
Metabolites and products for the reactions of liver ketogenesis. (**a**) Thiolase reaction. (**b**) HMG-CoA Synthase reaction. (**c**) HMG-CoA Lyase reaction. (**d**) β-hydroxybutyrate Dehydrogenase reaction. (**e**) Acetoacetate Decarboxylase reaction.

**Figure 2 metabolites-13-00803-f002:**
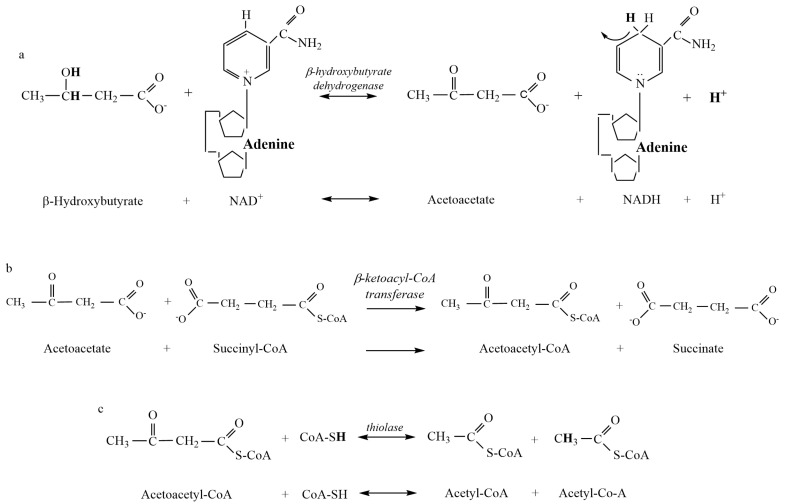
Metabolites and products for the reactions of ketolysis. (**a**) β-hydroxybutyrate Dehydrogenase reaction. (**b**) β-ketoacyl-CoA Transferase reaction. (**c**) Thiolase reaction. There is currently insufficient biochemical evidence and data to present the metabolic clearance of acetone as a component of ketolysis (see Section 4).

**Figure 3 metabolites-13-00803-f003:**
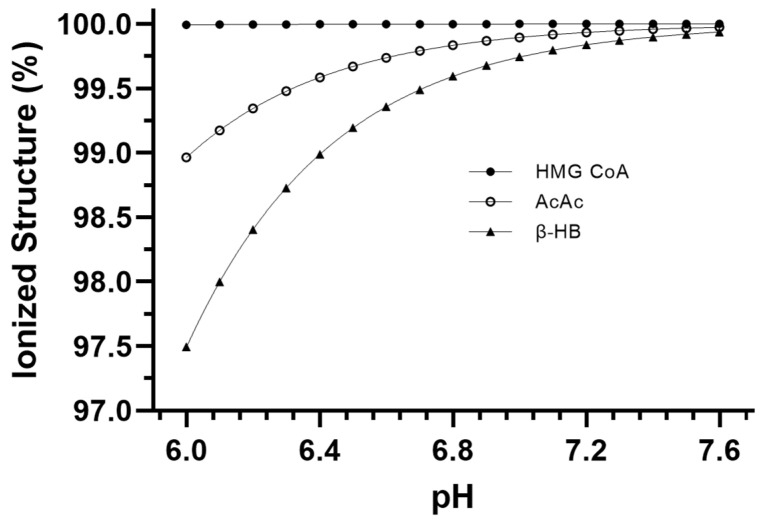
Relative ionization data for the main ionizable metabolites of ketogenesis and ketolysis across the physiological pH range.

**Table 1 metabolites-13-00803-t001:** The acid dissociation constants for the metabolites and related ionic complexes of the reactions of ketogenesis and ketone body oxidation, referenced to 24.85 °C and ionic strength of 0.1 mmol·L^−1^. Succinate (a) and (b) are used to differentiate between the two ionizable carboxylic acid groups [11]. na = not applicable due to the absence of pH-dependent ionizable groups.

Metabolite	pK_d_ *
Acetyl-CoA	na
Acetylacetyl-CoA	na
HMG-CoA	1.86
Acetoacetate	4.02
Acetone	na
β-hydroxybutyrate	4.41
Succinyl-CoA	3.87
Succinate (a)	3.55
Succinate (b)	5.69
Co-Enzyme A	na

* pK_a_ = pK_d._

**Table 2 metabolites-13-00803-t002:** An example of the H^+^ exchange calculation results for β-HB Dehydrogenase reaction of ketogenesis.

	AcAc	NADH	H^+^	β-HB	NAD^+^	H^+^ Coeff
K_d_	10,471.28	0		25,703.96	0	
pH						
6	0.0104	0	1	0.0251	0	1.0147
6.1	0.0082	0	1	0.0200	0	1.0118
6.2	0.0066	0	1	0.0160	0	1.0094
6.3	0.0052	0	1	0.0127	0	1.0075
6.4	0.0042	0	1	0.0101	0	1.0060
6.5	0.0033	0	1	0.0081	0	1.0048
6.6	0.0026	0	1	0.0064	0	1.0038
6.7	0.0021	0	1	0.0051	0	1.0030
6.8	0.0017	0	1	0.0041	0	1.0024
6.9	0.0013	0	1	0.0032	0	1.0019
7	0.0010	0	1	0.0026	0	1.0015
7.1	0.0008	0	1	0.0020	0	1.0012
7.2	0.0007	0	1	0.0016	0	1.0010
7.3	0.0005	0	1	0.0013	0	1.0008
7.4	0.0004	0	1	0.0010	0	1.0006
7.5	0.0003	0	1	0.0008	0	1.0005
7.6	0.0003	0	1	0.0006	0	1.0004

K_d_ = dissociation constant; H^+^ column = single covalent H^+^ uptake. See Figure 1d for the chemical structures of this reaction.

**Table 3 metabolites-13-00803-t003:** H^+^ coefficient data for reactions of ketogenesis and ketolysis, including net H^+^ coefficients for ketosis (sum of ketogenesis, blood transfer, and muscle or brain ketolysis) for different proportions of ketone bodies during ketogenesis. Note that for simplicity, only the main substrates and products are identified. See Figure 1 and Figure 2 for complete details of each reaction.

Reaction	Substrates		Products		H^+^ Coeff		
Ketogenesis					pH = 6	pH = 7	pH = 7.6
Thiolase	Acetyl CoA	Acetyl CoA	Acetoacetyl CoA	CoASH	0.0000	0.0000	0.0000
HMG-CoA Synthase	Acetoacetyl CoA	Acetyl CoA	HMG CoA	CoASH + H^+^	−0.9999	−1.0000	−1.0000
HMG-CoA Lyase	HMG CoA		Acetoacetate	Acetyl CoA	0.0103	0.0010	0.0003
β-Hydroxybutyrate Dehydrogenase	Acetoacetate	NADH + H^+^	β-Hydroxybutyrate	NAD^+^	1.0147	1.0015	1.0004
Acetoacetate Decarboxylase	Acetoacetate	H^+^	Acetone	CO_2_	0.9896	0.9990	0.9997
**Ketolysis**							
β-Hydroxybutyrate Dehydrogenase	β-Hydroxybutyrate	NAD^+^	Acetoacetate	NADH + H^+^	−1.0147	−1.0015	−1.0004
β-Ketoacyl-CoA Transferase	Acetoacetate	Succinyl-CoA	Acetoacetyl CoA	Succinate	0.3146	0.0453	0.0118
Thiolase	Acetoacetyl CoA	CoASH	Acetyl CoA	Acetyl CoA	0.0000	0.0000	0.0000

**Table 4 metabolites-13-00803-t004:** Net H^+^ exchange data for different proportionalities of ketone body cellular (or blood) concentrations based on liver ketogenesis (pH = 7.0) (β-HB, AcAc, Acetone), blood transfer (blood, pH = 7.4), and uptake into the brain (50%) (pH = 7.2) and muscle (50%) (pH = 7.1) for β-HB and AcAc.

Proportion *	Ketogenesis	Blood Transfer	Ketolysis	Net Summation
30:35:35	−0.3489	−0.0002	−0.1395	−0.4886
40:30:30	−0.2987	−0.0003	−0.1888	−0.4877
50:25:25	−0.2485	−0.0003	−0.2380	−0.4867
60:20:20	−0.1983	−0.0003	−0.2872	−0.4858
60:10:30	−0.0984	−0.0003	−0.2889	−0.3875
50:10:40	−0.0986	−0.0002	−0.2405	−0.3393

* for β-HB:AcAc:Acetone; Shaded row = reference condition most supported by prior research (see Section 2 and Section 4).

## Data Availability

Data available on request due to restrictions posed by institution intellectual property. The data presented in this study are available on request from the corresponding author.

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
