# Peer review of "The Computational Acid–Base Chemistry of Hepatic Ketoacidosis"

_metabolites, 2023, doi:10.3390/metabo13070803_

Round 1
Reviewer 1 Report
Referee report concerning the manuscript:
metabolites-2424435
The computational acid-base chemistry of hepatic ketoacidosis
by S. Torrens et al.
The authors studied sources of protons in the ketoacidosis process by using available experimental data, namely pKa values and pH values of various compartments. The authors demonstrated that proton release or uptake during ketogenesis and ketolysis are caused by covalent modification rather than acid dissociation. Moreover, they demonstrated that ketogenesis and ketolysis results in a net fractional proton release, that in turn depends on the levels of acetoacetate, beta-hydroxy-butyrate and acetone.
The manuscript describes a highly relevant topic, but it is not easy to read. The improvements I can suggest you prior publishing are as follows.
1) The manuscript is not easy to read. I would suggest the authors to add a couple of schemes/flowcharts. Moreover, pKa value of the relevant species should be added.
Most of the readers will benefit from that.
2) I would add a paragraph stating that pKa values depend on the environment and are difficult to measure and calculate.
See some work of Arieh Warshel and Al Mildvan. See for example:
Biochemistry. 2001 Feb 20;40(7):1984-95.
Biochemistry 1996, 35, 814-823
J. Phys. Chem. B 1997, 101, 4458-4472
J. Phys. Chem. B, 118 (2014) 4326-4332
Quote, comment!
3) I would mention enzyme catalyzed conversion of acetyl CoA to HMG-CoA. The enzyme is called
HMG-CoA reductase and was studied by Olaf Wiest and his group.
Biochemistry. 2012 Oct 9; 51(40): 7983–7995.
Quote, comment!
4) Kinetics of reactions involving ionizable groups may strongly depend on the pH value.
For dopamine autoxidation see
Front. Mol. NeuroSci. 11 (2018) article 467, doi: 10.3389/fnmol.2018.00467
Would approach like this apply also to the studied reactions?
Quote, comment!
--End of comments--
Author Response
Responses to reviewer comments are in italics.
Reviewer 1
The authors studied sources of protons in the ketoacidosis process by using available experimental data, namely pKa values and pH values of various compartments. The authors demonstrated that proton release or uptake during ketogenesis and ketolysis are caused by covalent modification rather than acid dissociation. Moreover, they demonstrated that ketogenesis and ketolysis results in a net fractional proton release, that in turn depends on the levels of acetoacetate, beta-hydroxy-butyrate and acetone.
The manuscript describes a highly relevant topic, but it is not easy to read. The improvements I can suggest you prior publishing are as follows.
Thank you for the review of this manuscript.
1) The manuscript is not easy to read. I would suggest the authors to add a couple of schemes/flowcharts. Moreover, pKa value of the relevant species should be added.
Most of the readers will benefit from that.
Note that we presented the pKd (=pKa) values of all pertinent metabolites in Table 1, page 5.
2) I would add a paragraph stating that pKa values depend on the environment and are difficult to measure and calculate.
See some work of Arieh Warshel and Al Mildvan. See for example:
Biochemistry. 2001 Feb 20;40(7):1984-95.
Biochemistry 1996, 35, 814-823
- Phys. Chem. B 1997, 101, 4458-4472
- Phys. Chem. B, 118 (2014) 4326-4332
While we understand the nature of your comment, we did not include such added writing and referencing for two main reasons ; 1) this issue is not pertinent to the manuscript as we used previously measured dissociation constants, and 2) the editing features of the journal are difficult for us to add new references and include added citations (more the former).
3) I would mention enzyme catalyzed conversion of acetyl CoA to HMG-CoA. The enzyme is called
HMG-CoA reductase and was studied by Olaf Wiest and his group.
Biochemistry. 2012 Oct 9; 51(40): 7983–7995.
You are mistaken here. HMG-CoA reductase catalyzes the conversion of HMG CoA to mevalonate and as such is not a reaction in ketogenesis.
4) Kinetics of reactions involving ionizable groups may strongly depend on the pH value.
For dopamine autoxidation see
Front. Mol. NeuroSci. 11 (2018) article 467, doi: 10.3389/fnmol.2018.00467
Would approach like this apply also to the studied reactions?
Yes, we recognize this and identified the pH dependence of our calculations within the Introduction and also Methods. See page 3, lines 143-145; page 5, line 163; page 6, lines 167-168, 172, 175, etc.
Reviewer 2 Report
The authors present a body of evidence for the source of hydrogen ions (H+) during ketoacidosis, disproving the traditional theory that ketone bodies are metabolic acids that cause systemic acidosis.
They used the dissociation constants and alpha equations for all ionizable metabolites involved in the chemical reactions of ketogenesis and ketolysis, which allowed identification of the sources and quantify added fractional (~) H+ exchange (~H+e).
The authors explain the mechanism charge-balanced chemical reactions and H+ of the production of ketone bodies (ketogenesis) and provides the dissociation constants for these reactions. The paper provides a number of recommendations for future research into metabolic explanations for systemic acidosis coinciding with ketosis.
I am very satisfied with the value of the work, which I consider attractive.
The abstract is well formulated.
The literature was well selected (using 25 of bibliographic references), and the critical analysis of the references is well done.
The working method involved competitive cation binding calculations for the proton balance for the reactions type ketoacidosis.
The graphs and tables are very well done, and the interpretation of the experimental results corresponds to the level agreed by the journal.
The calculation relations are well established, using the known equations.
Regarding the conclusions, they summarize very well the essence of the results obtained.
The bibliographic references (31) are rigorously selected from the current specialized literature.
The paper can be published.
Author Response
Responses to reviewer comments are in italics.
Reviewer 2
The authors present a body of evidence for the source of hydrogen ions (H+) during ketoacidosis, disproving the traditional theory that ketone bodies are metabolic acids that cause systemic acidosis.
They used the dissociation constants and alpha equations for all ionizable metabolites involved in the chemical reactions of ketogenesis and ketolysis, which allowed identification of the sources and quantify added fractional (~) H+ exchange (~H+e).
The authors explain the mechanism charge-balanced chemical reactions and H+ of the production of ketone bodies (ketogenesis) and provides the dissociation constants for these reactions. The paper provides a number of recommendations for future research into metabolic explanations for systemic acidosis coinciding with ketosis.
I am very satisfied with the value of the work, which I consider attractive.
The abstract is well formulated.
The literature was well selected (using 25 of bibliographic references), and the critical analysis of the references is well done.
The working method involved competitive cation binding calculations for the proton balance for the reactions type ketoacidosis.
The graphs and tables are very well done, and the interpretation of the experimental results corresponds to the level agreed by the journal.
The calculation relations are well established, using the known equations.
Regarding the conclusions, they summarize very well the essence of the results obtained.
The bibliographic references (31) are rigorously selected from the current specialized literature.
The paper can be published.
Thank you for the review of this manuscript.
Reviewer 3 Report
The main question of the research is whether metabolic acidosis related to the formation of ketone bodies. The formation of ketone bodies causes acidosis, because ketones are acidic. In addition, there are not many basic studies showing the H+ protonation pathway in intracellular ketone body generation. Therefore, research using computer algorithms with input data calculated H+ exchange in the responses of ketogenesis and primary ketolysis in tissues to answer the above question. The conclusion of the study is very closely related to the research question and the 4 objectives that the article sets out from the beginning. Tables and pictures of the content are informative enough on their own to make sense. I recommend to publish this manuscript with minor revision as follow:
Point 1: "The authors do not mention the SD value of the equation to confirm whether the calculation process is acceptable? How could we validate the outcome from the formula?"
Please check
Author Response
Response to reviewer comments in italics.
Reviewer 3
The main question of the research is whether metabolic acidosis related to the formation of ketone bodies. The formation of ketone bodies causes acidosis, because ketones are acidic. In addition, there are not many basic studies showing the H+ protonation pathway in intracellular ketone body generation. Therefore, research using computer algorithms with input data calculated H+ exchange in the responses of ketogenesis and primary ketolysis in tissues to answer the above question. The conclusion of the study is very closely related to the research question and the 4 objectives that the article sets out from the beginning. Tables and pictures of the content are informative enough on their own to make sense. I recommend to publish this manuscript with minor revision as follow:
Point 1: "The authors do not mention the SD value of the equation to confirm whether the calculation process is acceptable? How could we validate the outcome from the formula?"
Thank you for your review. Regarding the need for SD data to verify the validity of our Methods and resulting data, note that we used previously validated dissociation constants and alpha equations to compute pH specific H+ exchange features for the metabolites and chemical reactions of interest. There is no variability in this data to compute a SD. Nevertheless, we have added a sentence to the Discussion to reveal the further need to replicate these findings in in-vitro preparations of metabolic pathways (see page 11, lines 377, 378).